# Persistent Idiopathic Facial Pain (PIFP) in Patients Referred to a Multidisciplinary Centre in Italy: A Retrospective Observational Study

**DOI:** 10.3390/jcm11133821

**Published:** 2022-07-01

**Authors:** Vittorio Schweiger, Riccardo Nocini, Daniele De Santis, Pasquale Procacci, Giovanni Zanette, Erica Secchettin, Giovanna Del Balzo, Andrea Fior, Alvise Martini, Marta Nizzero, Katia Donadello, Gabriele Finco, Leonardo Gottin, Enrico Polati

**Affiliations:** 1Anesthesiology, Intensive Care and Pain Therapy Center, Department of Surgery, Dentistry, Paediatrics and Gynaecology, University of Verona, 37124 Verona, Italy; erica.secchettin@univr.it (E.S.); alvise.martini@univr.it (A.M.); marta.nizzero@gmail.com (M.N.); katia.donadello@univr.it (K.D.); leonardo.gottin@univr.it (L.G.); enrico.polati@univr.it (E.P.); 2Head and Neck Department, Department of Surgery, Dentistry, Paediatrics and Gynaecology, University of Verona, 37124 Verona, Italy; riccardo.nocini@univr.it (R.N.); daniele.desantis@univr.it (D.D.S.); pasquale.procacci@univr.it (P.P.); andrea.fior@aovr.veneto.it (A.F.); 3Clinical Psychology, Department of Neuroscience, Biomedicine and Movement, University of Verona, 37124 Verona, Italy; giovanni.zanette@univr.it; 4Department of Medicine and Public Health, Section of Forensic Medicine, University of Verona, 37124 Verona, Italy; giovanna.delbalzo@univr.it; 5Department of Medical Sciences and Public Health, University of Cagliari, 09124 Cagliari, Italy; gabriele.finco@unica.it

**Keywords:** persistent idiopathic facial pain, atypical facial pain, post-traumatic trigeminal neuropathy, classification

## Abstract

Background: Persistent Idiopathic Facial Pain (PIFP), previously named Atypical Facial Pain (AFP) is a poorly understood condition, often diagnosed after several inconclusive investigations. The aim of this retrospective study was to evaluate the demographic and clinical characteristics of patients with PIFP referred to a Facial Pain Center. Methods: Between May 2011 and September 2014, data on 41 PIFP patients were analyzed regarding temporal, topographical and descriptive pain features, including onset, localization, pain descriptors and intensity. Pharmacological pain treatments were also registered. Finally, the presence and type of previous minor oro-surgery procedures in the painful area were investigated. Results: Demographic and clinical characterization were similar to PIFP patients reported in literature. The presence of previous minor oro-surgery procedures in the painful area was reported in most of these patients, in particular endodontic treatments and tooth extractions. Conclusions: This retrospective analysis showed a high prevalence of minor oro-surgery procedures in our population, while its role in PIFP pathophysiology remains unknown. A new classification of PIFP built around the main discriminant factor of presence of these procedures in the painful area could be considered while available data were still insufficient to define specific diagnostic criteria.

## 1. Introduction

Persistent Idiopathic Facial Pain (PIFP), previously termed Atypical Facial Pain (AFP), is a poorly understood painful condition often diagnosed after several inconclusive clinical investigations. According to the last International Classification of Headache Disorders ICHD third edition 2018, PIFP is defined as persistent facial and/or oral pain, with varying presentations but recurring daily for more than 2 h/day over more than 3 months, in the absence of clinical neurological deficit. Pain must be hardly localized, not following the distribution of a peripheral nerve, dull or aching or nagging in quality. Clinical neurological examination must be normal, and a dental cause must be excluded by appropriate investigations [1]. PIFP is a rare painful condition, with a lifetime prevalence, estimated in a population-based sample, to be 0.03% [2]. A relevant risk factor for developing PIFP is the female gender, with 40 years of mean age at pain onset [3]. The pathogenesis of PIFP is still unknown. While a neuropathic feature has been postulated, the absence of neurological lesions or diseases make this diagnosis unlikely according to the more recent neuropathic pain guidelines [4]. However, some patients with PIFP report a history of minor oro-surgical procedures such as teeth extraction or implant-supported rehabilitation in the painful area, with no signs of tissue or nerve damages. In these cases, it has been assumed that a minor injury to orofacial structures leads to the modification of neural tissue physiology, though the neurologic processes involved have not been clearly demonstrated [5,6,7]. The aim of this retrospective observational study was to evaluate the clinical characteristics of a population of patients diagnosed as PIFP, with a special focus on minor oro-surgical procedures in the painful area before pain onset.

## 2. Materials and Methods

### 2.1. Study Design

In this retrospective observational study, performed according to the STROBE statement [8], we analyzed data collected during clinical practice at our Multidisciplinary Facial Pain Center regarding patients aged ≥18 years diagnosed with PIFP between May 2011 and September 2014. All available data were collected from the standardized clinical record form (CRF) routinely used during daily practice.

### 2.2. Data Extraction

From CRF we extracted demographic and clinical data regarding temporal, topographical and descriptive features of patient’s pain, including onset, localization, type, and intensity. Pharmacological pain treatments at the time of evaluation were also registered. Moreover, presence and type of minor oro-surgical procedures in the painful area and their temporal relationship with pain onset were investigated.

### 2.3. Data Analysis

All the available data were collected and analyzed. Demographic, medical, and clinical characteristics were summarized by descriptive statistics. Data collected in this study were retrospectively analyzed using MedCalc Statistical Software version 14.8.1 (MedCalc Software bv, Ostend, Belgium, 2018; https://www.medcalc.org (accessed on 18 May 2022)).

### 2.4. Ethics

All the study procedures were found to be in accordance with the Helsinki Declaration of 1975/83. The study was approved by the local ethical committee (1751CESC).

## 3. Results

Between May 2011 and September 2014, a diagnosis of PIFP was given to 41 patients.

A significant prevalence of female gender was highlighted (31 patients, 75.6%). The mean age was 53.8 years (±13.68), with no significant gender difference. Complete demographic characterization is summarized in Table 1.

Most patients reported episodic pain feature (33 patients, 80.48%), with at least one daily episode in 25 patients (60.9%). Eight patients (19.5%) reported continuous pain, sometimes with intermittent exacerbations. The most reported pain descriptors, often in association, were burning (46.3%), lancinating (41.5%), electric shock-like (36.6%), pulsating (31.7%) and gnawing (24.4%). In four patients (9.8%), the pain was described generically as “aching”. The mean daily pain intensity assessed by NRS scale (0–10) was 6.35 (±2.54). Pain characterization is summarized in Table 2.

Pain location was unilateral in 27 patients (65.8%) and bilateral in 14 patients (34.1%). In patients with unilateral pain, location was reported in maxillary (16 patients, 59.2%), mandibular (7 patients, 25.9%) or in frontal/periocular region (4 patients, 14.8%). Intraoral pain was reported by 32 patients (78.04%), while extraoral and intra/extra-oral pain was reported in six (14.63%) and in three patients (7.31%), respectively. Pain location is summarized in Table 3.

In most patients (32 patients, 78%), the pain disappears during night sleep. Regarding neurological assessment, none of the 41 PIFP patients reported neurological impairment such as hyperalgesia, allodynia, paresthesia, or anesthesia in the painful area. Active oro-facial tissue modifications were excluded by the maxillo-facial evaluation. Regarding pharmacological treatment at the time of evaluation, the majority of patients were undergoing NSAIDs treatment (37 patients, 90.24%) alone or in combination with antibiotics (13 patients, 35.1%). Anticonvulsant drugs were administered in four patients (10.8%), weak or strong opioids in two patients (5.4%), tricyclic antidepressants (TCA) in two patients (5.4%). Only four patients (9.7%) were drug-free at the time of evaluation. Pharmacological treatment is summarized in Table 4. The presence of minor oro-surgical procedures in the painful area before pain onset was registered in 34 patients (83%). These patients were predominantly female (26 patients, 76.4%) with a mean age of 51 years. The procedures reported were endodontic treatments (12 patients, 44.4%), tooth extractions (7 patients, 25.9%), apicectomy (4 patients, 14.8%), implant-supported rehabilitations (4 patients, 14.8%), dental prosthetic treatments (2 patients, 7.4%), dental conservative cares (2 patients, 7.4%), mandibular cyst enucleation (1 patient, 2.9%), intraoral mucosal biopsy (1 patient, 2.9%) and minor procedural trauma (1 patient, 2.9%). The majority of these procedures (65.9%) were performed in the superior maxillary region. Pain onset was simultaneous with the procedure (9 patients, 26.4%), within 3 months (16 patients, 47%) or after 3 months (9 patients, 26.4%). The time lapse between procedure and pain onset was significantly related with the type of procedure. In particular, tooth extractions and endodontic treatments showed a longer time interval (3 months or more) compared to other procedures (*p* = 0.031). Patients with previous minor oro-surgical procedures reported predominantly episodic pain (26 patients, 76.4%). The duration of daily pain episode was reported as a few seconds or minutes (5 patients, 19.2%) or hours (21 patients, 80.7%). Patients with continuous daily pain (8 patients, 23.5%) also reported transient intermittent pain exacerbations. In 27 patients (79.41%), pain was described with a unilateral location. In most of these patients (32 patients, 94.1%), pain disappeared during night sleep. Oro-surgical procedures are summarized in Table 5. The presence of minor oro-surgical procedures was not reported by seven patients (17.07%). All these patients were female and had referred bilateral facial pain.

## 4. Discussion

This retrospective observational study on a population of 41 patients diagnosed with Persistent Idiopathic Facial Pain (PIFP) showed a prevalence of female gender, with a main age of 53 years, a prevalence of episodic daily pain, and heterogeneity in pain-type descriptors, with a slight prevalence of burning and lancinating pain and a more frequent unilateral pain location in superior maxilla. Moreover, the clinical evaluation showed the complete absence of any kind of neurological or tissue impairment in the painful area. Most of the patients reported pain freedom during night sleep. The pharmacological pain treatment at the time of evaluation showed a prevalent use of NSAIDs and/or antibiotics while other analgesics, commonly used in chronic pain, were administered only in a minority of patients. Similar results were reported in other epidemiological observations, where a prevalence of female gender and a mean age of 40 years at pain onset were evidenced [9]. The analysis of clinical records of our small series showed the presence of minor oro-surgical procedures in painful area before pain onset in 34 patients, who were predominantly female, with a mean age of 51 years and unilateral pain location. Most of the aforementioned procedures were performed in the superior maxillary region. Tooth extractions and endodontic treatments showed a longer time interval between procedure and pain onset (≥3 months) compared to all the other interventions. In most of these patients, pain disappeared during night sleep. According to previous observations, ppatients undergoing minor oro-surgical procedures such as teeth extractions or implant rehabilitations may develop chronic facial pain without detectable persistent tissue damage or nerve lesions [10]. For this clinical entity, the label of Persistent Idiopathic Facial Pain (PIFP) is sometime used, while in several cases the consulted physician does not perform a conclusive diagnosis and refers the patient to other specialists to perform further evaluations. This process often extends the temporal interval between diagnosis and treatment [11]. The role of oro-surgical procedures in PIFP pathophysiology is still unknown. To date, the ICHD classification does not differentiate PIFP patients with or without a previous minor oro-surgical procedure in the painful area, while it has been hypothesized that these patients may suffer from a subform of PIFP or from a different type of PPTN (Painful Post-Traumatic Trigeminal Neuropathy), even though it lacks of precise characterization to propose specific diagnostic criteria and a separate classification [1]. However, it has been assumed that a traumatic injury to orofacial structures related to minor oro-surgical procedures can lead to a modifications of neural tissue physiology. Several observations in the literature have postulated the hypothesis of a neuropathic origin of PIFP based on specific neurophysiological findings. In particular, the blink reflex evaluation showed neurophysiological signs of neuropathy compatible with subclinical neuropathic pain in 15–23% of PIFP patients while application of QST (Quantitative Sensory Testing) showed sensory abnormality in small fibers in 55% of patients [11]. Moreover, alterations have been highlighted also at the central level in the nociceptive system. Increased neuronal excitability at the brainstem level, disturbed inhibitory function of the prefrontal cortex, and alterations in the dopamine systems associated with either/both pain transmission and its modulation were reported in PIFP patients [12]. A recent study on PIFP patients using cortical excitability measurements by transcranial magnetic stimulation applied to the cortical representation of the masseter muscle of both hemispheres showed changes in intracortical modulation involving GABAergic mechanisms, which may be related to certain aspects of the pathophysiology of this chronic pain condition [12]. Significant disturbances in somatosensory function, also called “central sensitization”, may explain the generalized dysfunction of the nociceptive system with imbalance between descending inhibitory and facilitatory mechanisms or deficiencies related to diffuse noxious control system [13]. Finally, some observations postulate that PIFP may involve a disproportionate response to mild injury, as occurs in patients with CRPS (Complex Regional Pain Syndrome), affected by traumatic injury and neuropathic response [14]. Conversely, as evidenced in our small series, some PIFP patients did not show the presence of a previous minor oro-surgical procedure in the painful area. The complete absence of triggering factors in some PIFP patients was observed also in other studies and the possibility of neuropathic involvement has been investigated. In fact, these PIFP patients showed unchanged somatotopy of the somatosensory cortex and inconsistent changes in the blink reflex, indicating no significant alterations in the trigeminal somatosensory pathways. Moreover, the QST profile, other than thresholds for warm and heat pain, is not significantly different to that in controls. In these patients, pain pathophysiology is completely unknown and even minor nerve damage can be excluded because of the absence of traumatic triggering factors. These findings suggest the possibility of two subtypes of PIFP, neuropathic and non-neuropathic; however, this dichotomy requires more data collection and analysis [14]. However, due to a small number of patients and the heterogeneity of pain features, our data were insufficient to put forward other defined diagnostic criteria for the two subgroups, except for the presence of minor oro-surgery procedures in the painful area in the patients’ clinical history and the absence of neurological impairment at the clinical evaluation.

## 5. Conclusions

Even though it requires further data collection, this small retrospective observation provided further evidence of a possible distinction between PIFP patients based on the presence of oro-surgery procedures in the painful area prior to pain onset. A new classification based on the presence of these procedures could represent a consistent tool for all specialists involved in diagnosis and treatment of this rare form of pain disorder. Lastly, our data collection could represent a useful tool for future research purposes.

## Figures and Tables

**Table 1 jcm-11-03821-t001:** Demographic characterization of PIFP population (41 patients).

Assessed for Evaluation	41
Gender, n (%)	
Female	31 (75.6%)
Male	10 (24.4%)
Age, year, mean (SD)	53.8 (±13.68)
TBPOE (years)	2.68
Smokers (%)	10 (24.4%)
Alcohol abuse (%)	5 (12.2 %)
Educational level, n (%)	
PSLC	17 (41.5%)
HS/U	24 (58.5%)
Comorbidity, n (%)	
Cardiovascular	16 (39%)
Gastroenterological	15 (36.5%)
Neurological/muscular-skeletal	6 (14.6%)
Genitourinary	6 (14.6%)
FMS	1 (2.4%)

TBPOE = Time between pain onset and evaluation; PSLC = Primary school leaving certificate; HS/U = High school or university; FMS = Fibromyalgia Syndrome.

**Table 2 jcm-11-03821-t002:** Pain characterization of PIFP population (41 patients).

Assessed for Evaluation	41
Main daily pain intensity, NRS (0–10)	6.35
Temporal pain characteristics	
Episodic pain, n (%)	33 (80.4%)
Continuous pain n (%)	8 (19.5%)
Pain type descriptors *	
Burning (%)	46.3%
Lancinating (%)	41.5%
Electric shock-like (%)	36.6%
Pulsating (%)	31.7%
Gnawing (%)	24.4%
Aching (%)	9.8%

NRS = Numerical Rating Scale; * alone or in combination.

**Table 3 jcm-11-03821-t003:** Pain topography of PIFP population (41 patients).

Assessed for Evaluation	41
Facial pain location	
Unilateral, n (%)	27 (65.8%)
Bilateral, n (%)	14 (34.1%)
Unilateral pain location	
Maxillary, n (%)	16 (59.2%)
Mandibular, n (%)	7 (25.9%)
Frontal/periocular, n (%)	4 (14.8%)
Bilateral pain location	
Single location, n, (%)	2 (14.2%)
Multiple location, n (%)	12 (85.7%)
Site of pain	
Intra-oral, n (%)	32 (78%)
Extra-oral, n (%)	6 (14.6%)
Intra and extra-oral, n (%)	3 (7.31%)

**Table 4 jcm-11-03821-t004:** Pharmacological treatments in the PIFP population (41 patients).

Assessed for Evaluation	41
Pharmacological pain treatments (alone or in combination)	
NSAIDs, n (%)	37 (90.24%)
Antibiotics, n (%)	13 (35.1%)
Anticonvulsants, n (%)	4 (10.8%)
Opioids, n (%)	2 (5.4%)
TCA, n (%)	2 (5.4%)
No treatment, n (%)	4 (9.7%)

NSAIDs = Non Steroidal Anti Inflammatory Drugs; TCA = Tricyclic antidepressant.

**Table 5 jcm-11-03821-t005:** Minor oro-surgical procedures in the PIFP population (34 patients).

Assessed for Evaluation	34
**Type of minor oro-surgical procedure**	
Endodontic treatment, n (%)	12 (44.4%)
Tooth extraction, n (%)	7 (25.9%)
Apicectomy, n (%)	4 (14.8%)
Dental prosthetic, n (%)	2 (7.4%)
Conservative care, n (%)	2 (7.4%)
Implant supported rehabilitation, n (%)	4 (14.8)
Small mucosal biopsy, n (%)	1 (2.9%)
Mandibular small cyst enucleation, n (%)	1 (2.9%)
Minor procedural trauma, n (%)	1 (2.9%)
**Location of procedure**	
Superior maxilla (%)	65.9%
Mandibular (%)	34.1%
**Pain onset after procedure**	
Simultaneous, n (%)	9 (26.4%)
Within 3 months, n (%)	16 (47%)
After 3 months, n (%)	9 (26.4%)

## Data Availability

All data related to the study are stored into the Institutional University Registry.

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
