# Peer review of "Persistent Idiopathic Facial Pain (PIFP) in Patients Referred to a Multidisciplinary Centre in Italy: A Retrospective Observational Study"

_jcm, 2022, doi:10.3390/jcm11133821_

Round 1

Reviewer 1 Report

The topic is interesting but the manuscript is very weak.

I found the following major flaws:

1. Title, aim of the study and conclusions are not corresponded one to each other within abstract and manuscript body.

2. What kind of internationally accepted guidelines have been used for TMJ and masticatory muscles assessment?

3. There is a lack of precise diagnoses for orofacial conditions presented in section 3.1. The presented description is too general.

4. There is a lack of summarizing table presenting chief findings of the study.

5. Discussion is not related to the aim of the study.

6. Authors have to report the study in accordance to the STROBE Statement which is international standard for reporting observational studies.

Authors wrote mainly about past and present criteria of PIFP in accordance to different guidelines and classifications but it is not a response on the aim of the study. According to the Authors the aim of this retrospective observational study was to evaluate the demographic and clinical        characteristics of a population of patients with chronic non-cancer orofacial pain.  Chronic non-cancer orofacial pain is not only limited to PIFP. Authors mixed  terms AO, AFP, PIFP with no logic scheme in different section of the paper. Whole manuscript is misleading and confusing.

I recommend to clarify and unify title, the aim, text of manuscript and conclusions. Currently it is very chaotic.

Author Response

Dear Reviewer,

we thank you for your timely and careful comments on our manuscript. In relation to your requests for content improvement, we would like to point out the following changes:

  • Title, aim of the study and conclusions are now correspondent within abstract and manuscript body
  • Data analysis was limited to the 41 patients diagnosed as PIFP in the CRF, without considering other facial pain diagnoses that are not of interest for the purpose of our observation. Other classifications that may be confusing have been eliminated, except for the acronym PPTN (post traumatic trigeminal neuralgia) as some authors consider the PIFP a subform of this entity.
  • Discussion is now related to the aim of the study
  • The STROBE statement was reported as standard method of our retrospective observation
  • Main observations are reported in the 5 tables.

We hope that these changes will allow a better understanding of our small experience

Reviewer 2 Report

The authors assessed persistent idiopathic facial pain (pifp) in patients referred to a multidisciplinary center in Italy: by an observational study. The main question addressed by the research was the determination of demographic and clinical characteristics of a population of patients with chronic non-cancer orofacial pain labeled as AFP and referring to a Multidisciplinary Facial Pain Center in Northern Italy. It was relevant and interesting. Although this is an interesting topic, it is only a descriptive study. This paper's main problem is that the sample size is small to draw some important and reliable conclusions. This is a preliminary study and should be regarded as a short communication.  The text was clear and easy to read. The conclusions were consistent with the evidence and arguments presented

Author Response

Dear Reviewer,

we agree that our manuscript referred to a small patients population but the PIFP disease is very rare condition. In our opinion, this observation may be useful for future studies and further classifications.

We hope that these changes and considerations will allow a better understanding of our small experience

Round 2

Reviewer 1 Report

The manuscript has been significantly improved. I don't have further comments.